# The Sedentary Lifestyle and Masticatory Dysfunction: Time to Review the Contribution to Age-Associated Cognitive Decline and Astrocyte Morphotypes in the Dentate Gyrus

**DOI:** 10.3390/ijms23116342

**Published:** 2022-06-06

**Authors:** Fabíola de Carvalho Chaves de Siqueira Mendes, Marina Negrão Frota de Almeida, Manoela Falsoni, Marcia Lorena Ferreira Andrade, André Pinheiro Gurgel Felício, Luisa Taynah Vasconcelos Barbosa da Paixão, Fábio Leite do Amaral Júnior, Daniel Clive Anthony, Dora Brites, Cristovam Wanderley Picanço Diniz, Marcia Consentino Kronka Sosthenes

**Affiliations:** 1Laboratório de Investigações em Neurodegeneração e Infecção, Instituto de Ciências Biológicas, Hospital Universitário João de Barros Barreto, Universidade Federal do Pará, Belém 66073-005, PA, Brazil; faesdam@yahoo.com.br (F.d.C.C.d.S.M.); marina_frota@hotmail.com (M.N.F.d.A.); manufalsoni@hotmail.com (M.F.); azulbx@hotmail.com (M.L.F.A.); agurgelfelicio@hotmail.com (A.P.G.F.); luisatpaixao@yahoo.com.br (L.T.V.B.d.P.); fabio.leite.amaral.jr@gmail.com (F.L.d.A.J.); cwpdiniz@gmail.com (C.W.P.D.); 2Curso de Medicina, Centro Universitário do Estado do Pará, Belém 66613-903, PA, Brazil; 3Laboratory of Experimental Neuropathology, Department of Pharmacology, University of Oxford, Oxford OX1 3QT, UK; daniel.anthony@pharm.ox.ac.uk; 4Research Institute for Medicines (iMed.ULisboa), Faculty of Pharmacy, Universidade de Lisboa, 1649-004 Lisbon, Portugal; dbrites@ff.ulisboa.pt; 5Department of Pharmaceutical Sciences and Medicines, Faculty of Pharmacy, Universidade de Lisboa, 1649-004 Lisbon, Portugal

**Keywords:** mastication, environment, aging, cognitive decline, astrocyte morphometry, dentate gyrus

## Abstract

As aging and cognitive decline progresses, the impact of a sedentary lifestyle on the appearance of environment-dependent cellular morphologies in the brain becomes more apparent. Sedentary living is also associated with poor oral health, which is known to correlate with the rate of cognitive decline. Here, we will review the evidence for the interplay between mastication and environmental enrichment and assess the impact of each on the structure of the brain. In previous studies, we explored the relationship between behavior and the morphological features of dentate gyrus glial fibrillary acidic protein (GFAP)-positive astrocytes during aging in contrasting environments and in the context of induced masticatory dysfunction. Hierarchical cluster and discriminant analysis of GFAP-positive astrocytes from the dentate gyrus molecular layer revealed that the proportion of AST1 (astrocyte arbors with greater complexity phenotype) and AST2 (lower complexity) are differentially affected by environment, aging and masticatory dysfunction, but the relationship is not straightforward. Here we re-evaluated our previous reconstructions by comparing dorsal and ventral astrocyte morphologies in the dentate gyrus, and we found that morphological complexity was the variable that contributed most to cluster formation across the experimental groups. In general, reducing masticatory activity increases astrocyte morphological complexity, and the effect is most marked in the ventral dentate gyrus, whereas the effect of environment was more marked in the dorsal dentate gyrus. All morphotypes retained their basic structural organization in intact tissue, suggesting that they are subtypes with a non-proliferative astrocyte profile. In summary, the increased complexity of astrocytes in situations where neuronal loss and behavioral deficits are present is counterintuitive, but highlights the need to better understand the role of the astrocyte in these conditions.

## 1. Introduction

Unhealthy brain aging and cognitive decline associate with a sedentary lifestyle and, at a cellular level, this is accompanied by astrocyte hypertrophy, myelin dysregulation, neurovascular dysfunction [1] and the impairment of neurogenesis [2]. Highly sedentary humans (≥8 h/day) display reduced hippocampal volumes and increased white matter (WM) hyperintensities [3,4] that are associated with accelerated cognitive, neuropsychiatric and functional decline [5]. In addition to the changes associated with a sedentary life, it has become clear that oral dysfunction is present in the same individuals and that this group feature is also associated with dementia or mild cognitive decline [6,7,8,9,10,11,12,13]. While it is not clear whether poor oral health predicts dementia, substantial data suggests that oral health declines as cognitive impairment and dementia progresses [7,12,14,15,16]. Furthermore, it has been demonstrated that masticatory exercise improves cognitive function in older adults [17] and thus the link between cognitive decline and masticatory dysfunction is now clear [8,18,19,20,21,22]. As loss of masticatory activity [8,11,12,13,18,19,20,21,22] and sedentary life style [23,24] are risk factors for age-related cognitive decline, there is a need to focus attention on those sub-populations that experience greater oral health deterioration or impairment of the stomatognathic system, and those having living sedentary lives.

Several experimental models of masticatory dysfunction have been explored to clarify the cellular and molecular mechanisms associated with memory impairment [25,26,27]. From these studies, we have learned that chewing maintains hippocampus-dependent cognitive function [18], and that age-related spatial memory deficits can be aggravated by a sedentary lifestyle and a reduction in masticatory activity [26,28,29,30,31]. In agreement with the findings described above, oral rehabilitation and environmental enrichment act in concert to restore spatial memory decline in aged mice [31]. In rat models of occlusal disharmony, amyloid-β is increased in the hippocampus and this was also associated with cognitive dysfunction [32,33]. Studies in similar mouse models of occlusal disharmony report significant increases in the expression of interleukin-1β in the brain, which was later accompanied by the appearance of amyloid-β and hyperphosphorylated tau in the hippocampus, and the induction of learning and memory deficits [34].

At the cellular level, cognitive decline has been linked to neuroinflammation via the enhanced activation of astrocytes, oligodendrocytes, and microglia [35,36,37] and these events are underscored by the presence of specific molecular signatures in the aging brain [38,39,40].

As a function of environmental stimuli [41,42], age [43,44], or the presence of other pathology [45,46,47], astrocytes differentially respond to changes in the microenvironment of the brain, in both form and function [48,49]. For example, physical exercise induces astrocyte proliferation and morphological changes, which alters the interplay between astrocytes, microglia and neurons to enhance neuroplasticity [23,50,51,52]. A distinctive pattern of gene expression is also induced in regions of the brain that are activated by exercise [53]. An enriched environment also induces neuroplastic changes in the dorsoventral hippocampal regions [54,55], increasing BDNF levels, p-AKT and p-MAPK1/2 and preventing neuroplastic decline by increasing the formation of dendritic spines and new neurons [24]. 

In rodent models of dysfunctional mastication, induced either by tooth loss, raised bite or soft diet, cognitive decline is associated with differential effects on astrocytes in different areas and different layers within the same greater brain region [56,57,58,59]. Indeed, five transcriptionally distinct astrocyte subtypes have been found in the mouse hippocampus [60,61]. In aged brains, previous transcriptomic analysis has revealed that there is upregulation of reactive astrocyte genes [62], which includes the expression of genes for neuroinflammation, synapse elimination pathways, and decreased cholesterol synthesis enzymes [63]. These changes were accompanied by an increase of A1 reactive astrocytes, which are argued to release a neurotoxic factor that induces neuronal death and cognitive decline [62]. In addition, dysregulated astrocytes and astrogliosis with an increased expression of GFAP and cellular hypertrophy [64] have been shown to be associated with impaired memory function in late life [65].

From optogenetics and chemogenetics studies, in which astrocytes can be selectively manipulated, emergent data has provided evidence that astrocytes directly participate in cognition [66,67,68,69], and other behavioral functions, including sensorimotor behaviours [70,71], sleep [72], feeding [73], fear and anxiety [74,75], and this is associated with regulation of synapses and circuits (see [76,77] for recent reviews). These tools may provide, in the near future, greater understanding of functional contributions of astrocyte subtypes within different areas of the central nervous system.

Here, we have re-evaluated morphological data concerning GFAP-positive astrocytes in the dentate gyrus at different ages under the influence of contrasting environments and induced masticatory dysfunction. We concluded that the differential effects of these challenges on astrocyte morphological complexities may reflect the existing transcriptomic diversity of astrocyte dentate gyrus phenotypes, and that environment and masticatory activity interact to alter the spatial distribution and morphology of glial fibrillary acidic protein in aged astrocyte arbors.

## 2. Running, Experiencing Novelty, and Mastication to Learn Faster, Better Remember, and Enhance Individual Ethological Behavior

It is well known that long-term voluntary running improves learning and memory, by enchancing the strength of neuronal connections, through synaptic plasticity in the hippocampus [78,79], and increasing neurogenesis [55,80]. Continuous voluntary wheel running exercise also contributes to astrogenesis and the repopulation of microglia [81]. The voluntary running-enhanced plasticity seems to be mediated by the Notch1 signaling pathway [82] and brain-specific angiogenesis inhibitor 1 (BAI1) [83]. In the absence of exercise, short-term [84] and lifelong environmental enrichment are able to improve memory and postpone age-related cognitive decline [85]; but for rodents enriched cages usually combine elements for physical exercise and cognitive stimuli. Indeed, running wheel, toys, tunnels, bridges, ropes, stairs, which are replaced or displaced from time to time (1 or 2 weeks) [86,87], encourage locomotor and exploratory activity in these cages, whereas the absence of these elements in standard laboratory cages do not. 

The elements inside enriched cages provide novelty, visuo-spatial and somatomotor stimuli and social interaction, but the stimuli for neurogenesis and the release of neurotrophins originate from voluntary exercise [88]. Comparative effects of the elements provided by an enriched environment have enabled the disentanglement of the influence of novelty, social and physical activity and behavioral performance in hippocampal-dependent tasks. Indeed, well designed comparative studies demonstrated that running stimulates hippocampal neurogenesis, while a complex environment does not. A complex environment, and not running, increases depolarization-associated c-fos expression and reduces plasma corticosterone [89]. However, the combination of cognitive stimuli, social interaction, and physical exercise was found to be the most effective way to reduce neuropathological outcomes in a transgenic mouse model of cerebral amyloid angiopathy [90].

Innate behavioral and physiological programs ensure survival and must be flexible enough to cope with environmental changes and build adaptive responses [91,92]. The impoverished environment of standard laboratory housing is associated with reduced display of species typical behaviors, whereas enriched cages seem to enhance ethological natural behaviors and increase individualized behavior in mice [55,93]. Hiding behavior is a good example of the innate repertoire to avoid attack and predation and this is a species-specific response that may explain the tendency of a mouse to avoid open/lit areas and to spontaneously explore unfamiliar areas [94]. In an open arena, for example, this mouse behavior is readily recognized as a preference for the safety of the peripheral zone of the open field [95]. Another innate typical behavior is related to the detection and exploration of novelty. In general terms, novelty is defined as a new event with which partial or no previous experience has occurred [96], being classified respectively as contextual/spatial novelty or stimulus novelty [97]. Enriched cages provide periodic inanimate object novelty and complexity through alterations in the physical and social environment, and these elements enhance sensory, cognitive and physical stimulation [98]. Similarly, an enriched environment enhances spatial learning, reversal learning and memory through the balance of excitatory and inhibitory synaptic densities [99]. The exploration of novelty related to a social stimulus or object recognition in rodents is known to activate different neural circuits [96], which appear to be an evolutionary adaptive response to provide parallel processing for novelty.

Oral and cognitive health are interconnected [100] and the recovery of masticatory activity can prevent cognitive decline [101,102,103]. The use of dental human prostheses successfully reduces cognitive consequences of masticatory dysfunction [104]. In animal studies, the relation between decrease in masticatory activity, due to a soft diet [28,30,105] or tooth loss [106], and memory impairment have been previously demonstrated [107]. Similarly, occlusal disharmony induces spatial memory impairment [29,106,108,109,110] and chronic stress [111,112,113]. Coherently, mastication activity, as a stress-coping behavior [114], is associated with activation of the hypothalamic-pituitary-adrenal (HPA) axis and hippocampus [115]. 

Mice housed in standard cages have reduced physical fitness and impaired thermoregulation, which leads to decreased ethological behavior and welfare [116]. In addition, long-term powdered diet increases the spontaneous locomotor activity of mice and their social interaction or impulsive and anxiety-like behaviors in elevated-plus-maze tasks [117]. These changes are associated with significant modifications in dopaminergic/noradrenergic systems and γ-aminobutyric acid-ergic (GABAergic) mediations in the frontal cortex [118]. In contrast, chewing prevents stress-induced hippocampal long-term depression (LTD) formation and anxiety-related behaviors, while ameliorating stress-induced suppression of hippocampal long-term potentiation (LTP) [119] via histamine H1 receptor [119]. Indeed, gene expression after weaning varies as a function of soft (reduced masticatory activity) or chow (normal masticatory activity) diets. In this study, gene ontology analysis of differential expression in the thalamus showed that glutamate decarboxylase, GABA receptors and the vesicular GABA transporter were upregulated in the chow diet group, whereas dendritic spine morphogenesis was downregulated, with a significant reduction in the number of spines at the ventral posterolateral and posteromedial nucleus [120]. 

The hypothalamic paraventricular nucleus (PVN), a high order integration center between the neuroendocrine and autonomic nervous systems, is affected by chewing, which reduces the number of corticotropin releasing factor positive cells inhibiting the autonomic releasing of adrenaline and noradrenaline via locus coeruleus [120]. 

We previously examined, in an open field, the combined influences of contrasting environments and masticatory regimens on exploratory and locomotor activity and found that all mice, independent of the masticatory condition, environment, or age, exhibited a similar temporal organization of their spatial horizontal exploratory activity in the open field task. However, aged mice living in life-long environmental enrichment and with normal masticatory activity showed reduced tendency to avoid open/lit spaces and that a contrasting diet regimen—a reduction or reduction/rehabilitation of mastication—showed differential effects at different ages. The combined effects of aging, environmental impoverishment and reduction in masticatory activity affect the innate behavioral repertoire of mice to explore novel environments and to assess risk [121].

## 3. Enriched Environment and Masticatory Rehabilitation to Prevent Synaptic Dysfunction Associated with Age-Related Cognitive Decline 

As lifelong environmental enrichment [85], physical exercise [122] and normal masticatory activity [100] may prevent cognitive decline, we re-evaluated our findings of combined induced masticatory dysfunction, oral rehabilitation and enriched environment on spatial learning and memory of aged mice. Data were obtained from three different conditions: (1) to mimic sedentary and active lifestyles we raised mice in standard or in enriched cages; (2) to induce masticatory dysfunction we used soft diet and compared with hard diet; and, (3) to measure age effects on spatial learning and memory, we compared mature adult (6M old) with aged (18M old) mice. Figure 1 is a schematic representation of the experimental approach.

In animal models all the approaches that have been used to induce masticatory dysfunction (soft diet feeding, molar extraction and bite raising) are associated with impairment of spatial learning and memory, a reduction of the number of hippocampal pyramidal neurons, the downregulation of brain derived neurotrophic factor, decreased synaptic activity, impaired neurogenesis in dentate gyrus and increased glial cell proliferation, which seem to be dose-dependent through the reduction of chewing-related stimuli (see [27,123] for systematic reviews). 

The synaptic changes in form, function, and plasticity associated with learning and memory formation are interrelated in the hippocampus [124,125,126]. As the hippocampal circuits mature, the establishment of synaptic reinforcement occurs in association with lasting structural changes and long-term potentiation (LTP). The intense synaptogenesis in the developmental period is replaced by an increase and clustering of mature synapses [127] and these synaptic rearrangements are selective and strengthen the circuits related to the task being learned [128].

A re-evaluation of the findings related to the performance of aged mice in the Morris water maze, and comparative effects of combined masticatory and environmental changes, are shown in Figure 2A–D. Figure 2A shows the influence of aging (3, 6 and 18 months) and masticatory reduction on mouse learning rates in the Morris water maze. Figure 2B shows the combined effects of environmental enrichment and masticatory activity rehabilitation in the recovery of spatial learning and memory, while Figure 2C,D demonstrate that this effect is not dependent on differential swimming speed, but is dependent on learning and memory. Taken together, the combined effects of these influences demonstrate that a reduction in masticatory activity (HD/SD) reduces learning rates at all ages, independent of the environment, and masticatory rehabilitation and environmental enrichment recover age-related memory impairments [30,31,59].

Functional magnetic resonance imaging studies have shown that when a comparison of the activity in the hippocampal subfields is made, the dentate gyrus (DG) is more active than the horn of Amon (CA1-CA2-CA3) and the subiculum, and that in both the coding process and information retrieval, the rostral (septal) pole is more active than the caudal (temporal) pole [129]. In fact, adult rats trained to remember the spatial location of an object, exhibited remodeling of synapses 6 h later in the molecular layer of the dorsal DG (septal) (DG-Mol) [130].

The entorhinal-to-dentate gyrus pathway is involved in memory formation carrying spatial and non-spatial information through the medial and lateral perforant excitatory pathways onto granule cells [131,132]. Astrocytes sense local synaptic transmission in the molecular layer of the dentate gyrus and control these inputs to the dentate granule cells at the presynaptic level [133,134,135]. 

Evidence has now emerged in rodents that the astrocyte is an essential mediator of learning and memory [135] and that astrocytic ephrin-B1 controls synapse formation in the hippocampus during learning and memory by regulating new dendritic spine formation and clustering on hippocampal neurons activated during memory recall [136]. Astrocytic processes encapsulate synapses allowing bidirectional communication with neurons [136] through G-protein-coupled receptors influencing learning and memory [137]. The activation of hippocampal astrocytes enhances synaptic potentiation and memory acquisition [66,138,139]. 

In the next section, we review our previous findings on learning and memory impairment induced by age, masticatory dysfunction and sedentary lifestyle and discuss the underlying mechanisms associated with differential effects on the morphological complexity of astrocytes at the molecular layer of the dentate gyrus.

## 4. Dentate Gyrus Astrocytes, Long Life Sedentary Lifestyle and Dysfunctional Mastication

It is known that physical exercise promotes morphological changes in astrocytes, and astrocytes may contribute to episodic memory function [42,140]. Astrocytic activation is necessary for synaptic plasticity and is sufficient to induce NMDA-dependent long-term potentiation in the hippocampus in a task-specific way, coupled with learning [138]. Since form precedes function, we explored the hypothesis that astrocyte morphological changes may reflect perforant pathway activity [127]. 

Here, we re-evaluated our findings of mouse age-related cognitive decline and used astrocyte morphological complexity to disentangle the multivariate morphological changes that were induced [59]. Our previous analysis of three-dimensional (3D) microscopic reconstruction of 1800 GFAP immunolabeled astrocytes from the dentate gyrus molecular layer, showed that after hierarchical cluster and discriminant analysis of 20 morphological features, two main astrocyte phenotypes could be identified. To this binary classification, we adopted the highest Euclidian distance between the two main clusters. Discriminant analysis suggested that morphological complexity was the morphological feature that most contributed to the cluster formation, so we named the two morphotypes AST1 and AST2, respectively, as a function of greater and lower mean values of morphological complexities of each. Figure 3 is a pictorial quantitative representation of age, environment, and masticatory activity influence on morphological complexity of AST1 and AST2 astrocytes. 

Morphological complexity definition in the present report was adapted to astrocyte morphology from the definition of morphological complexity of neuronal dendritic arbors described elsewhere [141] as follows:Complexity = [Sum of the terminal orders + Number of terminals] × [Total branch length/Number of primary branches](1)

In general, astrocyte arbors with the greater complexity phenotype (AST1) from an enriched environment, independent of masticatory regimen or age, showed thinner and more ramified branches than astrocytes from mice raised in an impoverished environment. This effect, however, is not readily recognized in astrocytes with the lower complexity phenotype (AST2). Thus, AST1 and AST2 morphological complexities are diversely affected by environment, aging and masticatory dysfunction, suggesting that astrocyte morphology does not respond linearly to these influences and that these morphotypes may have differential physiological roles.

Astro-glial morphological atrophy and loss of function seem to be part of neuropathological changes of the aging brain [142], and astrosenescence is characterized by loss of function and neuroinflammation, which seem to be central components to the mechanisms of age-related neurodegenerative disorders [35]. Astrocyte senescence is associated with an increased expression of glial fibrillary acid protein and vimentin [143], and aged astrocytes are associated with the releasing of chemokines, cytokines, and proteases [63,144]. Morphological [86] and metabolic astrocyte changes [145] also emerge as aging progresses and these changes can be aggravated by a sedentary lifestyle and masticatory dysregulation [30,31,146,147,148]. 

It has been suggested that astrocytes exhibit two main phenotypes associated with a proliferative profile surrounding areas of damaged tissues and a non-proliferative, but reactive, profile retaining basic structural organization and cell interactions in intact tissues [149]. All reconstructed astrocytes previously described [59] retained basic structural organization in intact tissue, suggesting that AST1 and AST2 phenotypes are indeed subtypes of a non-proliferative, reactive profile. Our findings suggest that astrocyte reactivity is not part of the neuropathological outcomes within the ageing brain and that they are influenced by masticatory dysfunction and sedentary lifestyle. Here, it is important to highlight that the term ‘reactive astrocyte’ is limited to astrocytes that undergo morphological, molecular, and functional changes in response to disease of the central nervous system, injury, or experimental damage [49,142]. For nomenclature, definitions, and future directions, see [49].

## 5. Differential Effects of Sedentary Lifestyle and Masticatory Dysfunction on Dorsal/Ventral Dentate Gyrus Morphological Phenotypes 

Although dorsal and ventral hippocampal regions show similar laminar and cellular organization, their connectivity to other brain regions are different [150,151,152,153,154]. They exhibit differential rates of neurogenesis and each displays a distinct pattern of neurotransmitter receptor distribution [155,156,157]. In addition, the septal/temporal divisions of the hippocampus exhibit significant differences in behavior-induced arc gene expression [158], distinct transcriptional and epigenetic effects in response to an enriched environment or physical activity [82,159], and distinct pathological responses throughout aging [160]. The dorsal hippocampus is associated with spatial memory and contextual information processing, while the ventral hippocampus is related to emotional behavior in association with fear, anxiety, and reward processing [161,162,163]. For example, small lesions in either the dorsal or ventral hippocampus generate distinct behavioral impairments in working memory and reference memory retrieval [164] and normal or abnormal neurogenesis along the septal/temporal hippocampal regions, which may be connected to mental health, neurological diseases [165,166] or affective disorders [167]. 

Astrocyte morphology is affected by aging [160,168,169,170,171,172], enriched environment [41,173] and masticatory dysfunctions [27], in a region- and subregion-specific manner. However, our previous astrocyte analysis related to masticatory dysfunction, age and sedentary lifestyle did not explore the septal-temporal (dorsoventral) hippocampal division. Therefore, we now re-analyzed our previous 3D morphometric reconstructions and compared dorsal and ventral astrocyte morphologies in the DG of the same individuals. 

We used a stereological sampling approach to select astrocytes from the molecular layer of dorsal and ventral dentate gyrus for three-dimensional reconstructions and the Hierarchical Ward’s Minimum Variance Clustering Method [174] was applied to variance-shrunk logarithmic values of multimodal morphometrical features to classify cells [175]. Following discriminant analysis, we found that morphological complexity was by far the variable that most contributed to cluster formation in most experimental groups. Indeed, 19 of 24 groups (10 in 12 and 9 in 12 experimental groups for dorsal and ventral dentate gyrus, respectively) shared morphological complexity as the variable that significantly contributed to cluster formation. The next variable shared by the experimental groups that contributed to cluster formation was the convex hull volume (9 in 24).

Figure 4 and Figure 5 show data revealing significant differences between the mean values of morphological complexity of astrocyte morphotypes suggested by hierarchical cluster and discriminant analysis. The designation of morphotypes 1-4 was based on their decrescent morphological complexity mean values with morphotype 1 and 4 corresponding to higher and lower mean values, respectively. 

Three main astrocyte morphotypes, exhibiting significant differences in morphological complexity mean values, were found in all experimental conditions. These phenotypes were differentially affected by dysfunctional mastication, sedentary lifestyle, and aging. A fourth astrocyte phenotype, with very low morphological complexity values, was found in 5 of 12 experimental conditions in the dorsal dentate gyrus and in only one instance the ventral dentate gyrus. Due to its asymmetric distribution in the experimental conditions and low occurrence, the comparative analysis of the 4th phenotype with the other phenotypes could not be undertaken and they were removed from our analysis. 

Data for the three main phenotypes from dorsal and ventral dentate gyrus are exhibited in Figure 6 and Figure 7, respectively. In general, the differential effects of diet regime on dorsal region were strong in AST1 and AST2 astrocytes from 6M-old mice from impoverished environment and in AST3 18M-old mice from enriched environment. Please note that environmental enrichment reversed all effects induced by diet regime on AST2 astrocytes. In general, the differential effects induced by environment and age on the mean values of morphological complexity of dorsal dentate gyrus astrocytes seem to be stronger than those induced by diet regime.

In contrast to astrocytes reconstructed from the molecular layer of dorsal dentate gyrus, the influence of the masticatory regime on the mean value of astrocyte morphological complexity was more conspicuous in the ventral region. Indeed, all chewing regimes, regardless of environment and age, influenced the mean value of morphological complexity of AST1 and AST2 morphotypes. Interestingly, the AST3 subtype in the ventral dentate gyrus was influenced by age and diet regime, but it was not influenced by environment. 

A previous report, limited to search for age influence on morphological complexity of GFAP astrocytes, demonstrated remarkable heterogeneity in the age-related changes in distinct subfields and along the dorsoventral axis of the hippocampus and in the entorhinal cortex of C57Bl6 mice [160]. These authors found that compared to 6-month-old mice the number of intersections, as a function of soma distance, increased significantly in dorsal dentate gyrus of 14-month-old mice, and the total sum of intersections, the number of processes and the total branch length followed a similar tendency, but no changes were observed in the ventral dentate gyrus. 

Recently [176], cyclic multiplex fluorescent immunohistochemistry was used to classify astrocytes morphologically in normal aging and Alzheimer’s Disease, and showed three main phenotypes of astrocytes: homeostatic, intermediate, and reactive. Reactive astrocytes and, to a lesser extent, intermediate astrocytes were associated with Alzheimer’s disease pathology. The intermediate astrocytes were suggested to represent a transitional state between reactive and homeostatic or to represent a resilience mechanism. These authors concluded that the classic binary “homeostatic vs. reactive” classification for astrocytes, but also relevant to microglia, may now include a third state that may represent gain or loss of function. Nevertheless, recent literature points out that astrocytes are heterogeneous and dynamic phenotypes with timing- and context-dependent states [74,149,177]. Based on Escartin et al. findings that “reactive astrocytes” encompass multiple potential states [49], our revisited findings of astrocyte morphological families did not include the designation of reactive astrocytes. In the future, morphological, molecular, and functional changes in response to ageing and pathological conditions in the surrounding tissue should be integrated to qualify astrocytes as reactive.

## 6. Concluding Remarks

Overall, we have reviewed the rapidly expanding literature on astrocyte behavior in homeostatic conditions to highlight how the cellular and molecular characteristics relate to behavior. In particular, we have focused on the cognitive decline aggravated by a sedentary lifestyle and by masticatory disorders by re-evaluating their influence on the morphology of the dentate gyrus astrocytes. We found that reduction of masticatory activity reduced learning rates at all ages, which was independent of the environment, and that masticatory rehabilitation and environmental enrichment recovered age-related memory impairments. These spatial learning and memory deficits, and their recovery by an enriched environment and masticatory rehabilitation, were associated with differential morphological changes of the dentate gyrus astrocytes. Unbiased selection of astrocytes for three-dimensional reconstruction from the external one-third of the molecular layer of the dentate gyrus, using a random and systematic stereological approach followed by hierarchical cluster and discriminant analysis, revealed the presence of four main astrocyte morphotypes. These morphological families, identified in the external third of the molecular layer of both dorsal and ventral dentate gyrus, were differentially affected by age, sedentary lifestyle, and masticatory dysfunction, which suggests that they have different physiological roles in homeostatic conditions. All the morphotypes retained basic structural organization in intact tissue, suggesting that they are indeed subtypes of a non-proliferative astrocyte profile. Future studies directly assessing astrocyte functions in the aging mouse brain should integrate molecular, cellular systemic and behavioral analysis to unravel the underlying mechanisms associated with aging cognitive decline aggravated by sedentary lifestyle and masticatory dysfunction. 

## Figures and Tables

**Figure 1 ijms-23-06342-f001:**
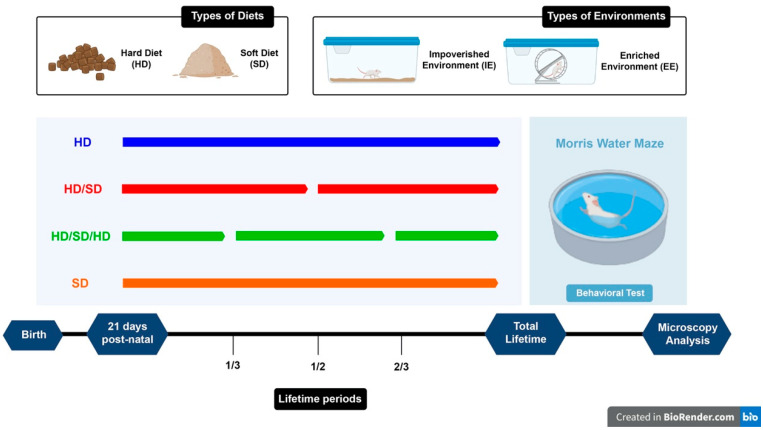
Experimental timeline. Female Swiss albino mice were fed under one of the following diet regimens: continuous hard (pellet) (HD), continuous soft (powder) diets (SD), sequences of hard and soft diet (HD/SD), as well as series of hard, soft, and hard diets (HD/SD/HD). Mice were maintained either in standard or enriched cages from 21st postnatal day onwards. Continuous or interrupted colored lines indicate continuous or interrupted diet regimens respectively: blue, red, green, and orange lines indicate HD, HD/SD, HD/SD/HD, and SD diet regimens respectively. The Morris water maze (MWM) behavioral tests are schematically represented on the right side of the scheme. Graphic representation of soft and hard diets, as well as standard and enriched cages are on the top.

**Figure 2 ijms-23-06342-f002:**
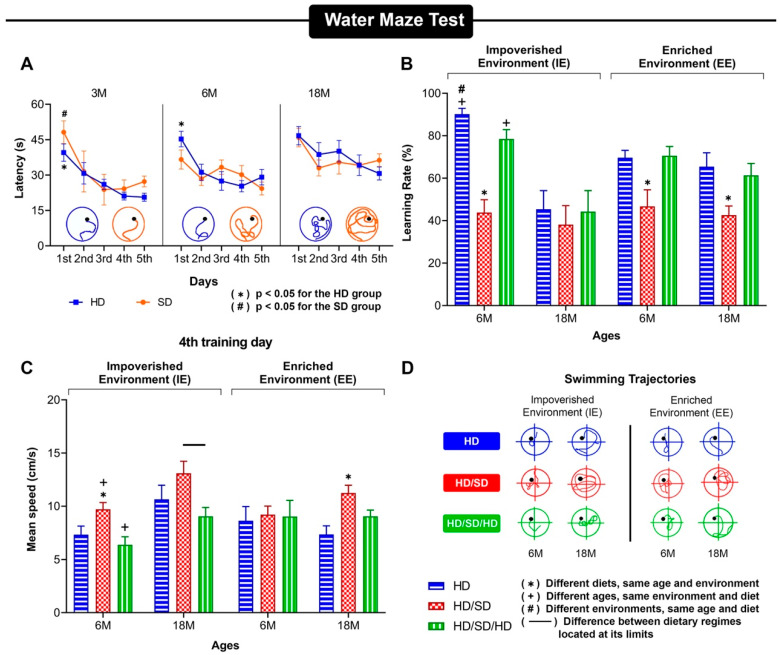
Influences of contrasting environments (impoverished and enriched), differential age and masticatory activity on Morris Water Maze spatial learning and memory performances in albino Swiss mice. (**A**) 3-, 6- and 18-month-old mice with correspondent mean values of escape latencies on five consecutive testing days. Blue- and orange-colored curves show progressive reduction in escape latency for hard diet (HD) and soft diet (SD) groups. Swimming tracking analysis below the curves shows that aged mice under SD exhibited longer trajectories than mice with HD at 6- and 18-month-old [30]. (**B**) Graphical representation of the influence of contrasting environments and masticatory activity rehabilitation in sequences of hard, soft, and hard diets (HD/SD/HD) on learning rate expressed as percentage 4th testing day values. Note that reduction of masticatory activity with sequences of hard and soft diet (HD/SD) reduced mouse learning rates at all ages independently of experienced environments, and this was not related to swimming speed (cm/s) (**C**). (**D**) Swimming trajectories for each group was selected based on the average distance closer to mean values of each group [31]. Results are expressed as mean ± standard error. M, month.

**Figure 3 ijms-23-06342-f003:**
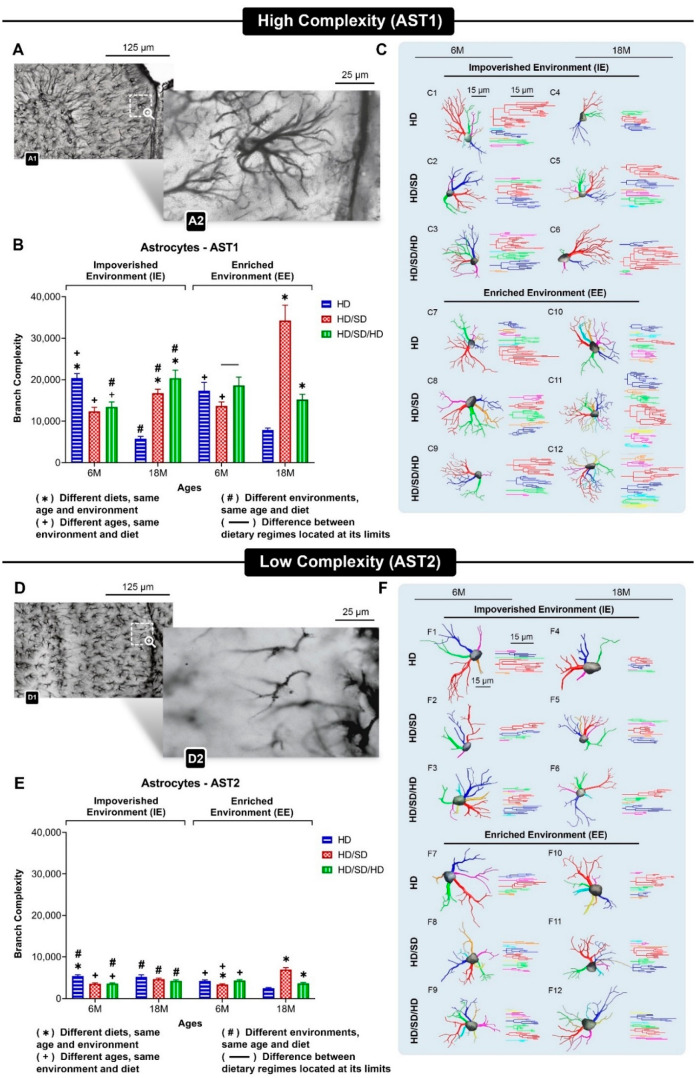
Three-dimensional reconstructions of astrocyte phenotypes (AST1 and AST2), under influence of different diet regimens (HD, HD/SD, and HD/SD/HD), environments (impoverished-IE vs. enriched-EE) and age (6M vs. 18M old). (**A**,**D**): Low (A1 and D1) and high (A2 and D2) power photomicrographs of GFAP-immunolabeled astrocytes to illustrate high (AST1) and low (AST2) morphological complexities of astrocytes from the external one third of molecular layer of mouse dentate gyrus. (**B**,**E**): Mean values of branch complexity and corresponding standard errors of astrocyte arbors to illustrate morphological differences between AST1 (**B**) and AST2 (**E**) for each experimental group. HD: hard diet/pellet food and SD: soft diet/powder food. (**C**,**F**): Three-dimensional reconstructions of the morphological phenotypes of astrocytes located in the outer 1/3 of the molecular layer of dentate gyrus with respective dendrograms. To choose the representative cell of each group, the distance matrix was used to obtain the sum of the distances of each cell in relation to all the others. Branches originating from the same parental trunk (primary branch) are shown with the same color. (**C**) AST1; (**F**) AST2. Dashed white squares identify the anatomical region from where photomicrographs of illustrated cells were taken. Scale bars: A1/D1 = 125 μm; A2/D2 = 25 μm. IE: impoverished environment; EE: enriched environment; 6M: six-month-old; 18M: eighteen-month-old; 9M: nine-month-old; HD: hard diet/pellet; SD: soft diet/powder food. AST1: astrocyte with high morphological complexity; AST2: astrocyte with low morphological complexity.

**Figure 4 ijms-23-06342-f004:**
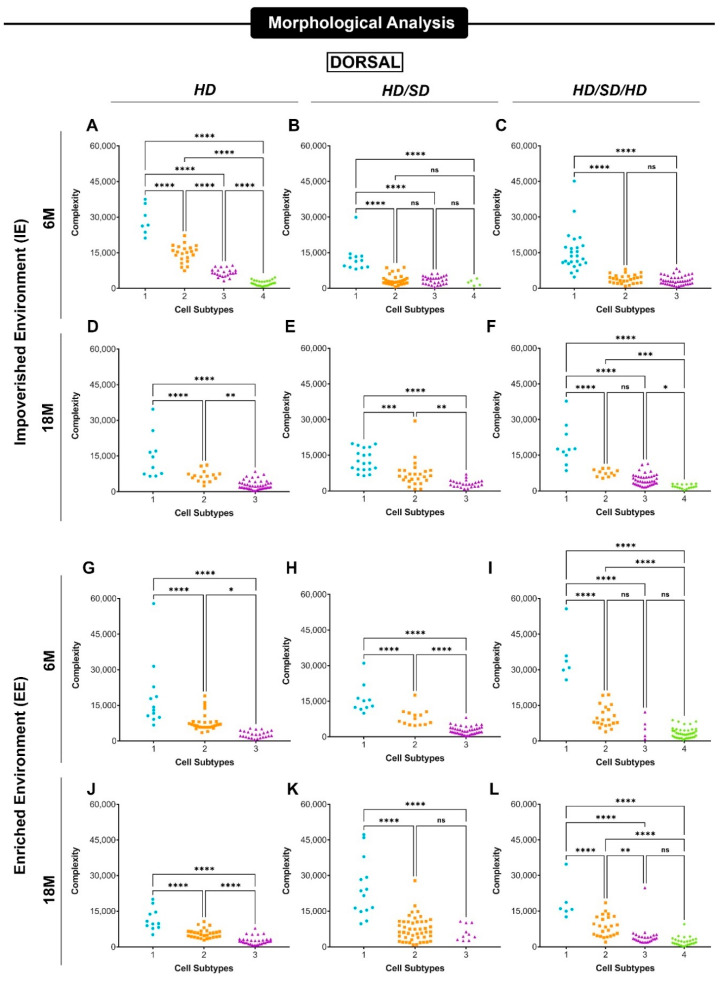
Mean values of astrocyte morphological phenotypes based on the cell arborization complexity at the external one-third of the molecular layer of the dorsal dentate gyrus. Impoverished environment—IE (**A**–**F**); Enriched environment—EE (**G**–**L**) age (6M—6-month-old; 18M—18-month-old) and masticatory regimen (HD, HD/SD, and HD/SD/HD). Morphological complexity is expressed as Mean/SD. Astrocyte morphotypes are indicated as 1–4 under each colored dataset. HD: hard diet/pellet; SD: soft diet/powder food; GFAP. * *p* < 0.05, ** *p* < 0.01, *** *p* < 0.001 and **** *p* < 0.0001; ns, not significant.

**Figure 5 ijms-23-06342-f005:**
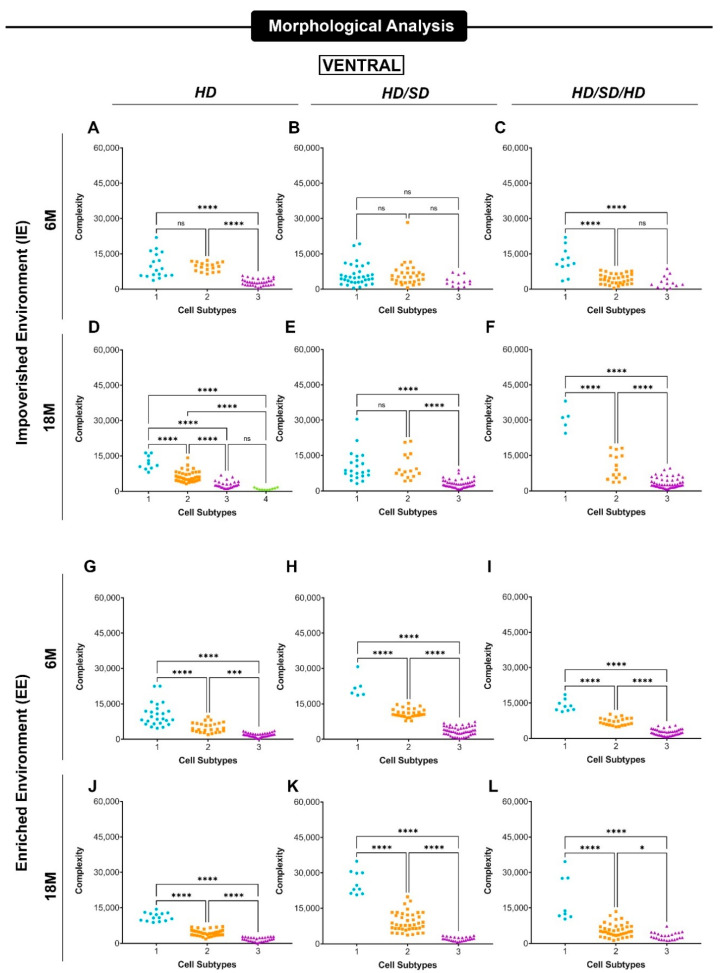
Mean values of astrocyte morphological phenotypes based on the cell arborization complexity at the external one-third of the molecular layer of the ventral dentate gyrus. Impoverished environment—IE (**A**–**F**); Enriched environment—EE (**G**–**L**), age (6M—6-month-old; 18M—18-month-old) and masticatory regimen (HD, HD/SD, and HD/SD/HD). Morphological complexity is expressed as Mean/SD. Astrocyte morphotypes are indicated as 1-4 under each colored dataset. HD: hard diet/pellet; SD: soft diet/powder food; * *p* < 0.05, *** *p* < 0.001 and **** *p* < 0.0001; ns, not significant.

**Figure 6 ijms-23-06342-f006:**
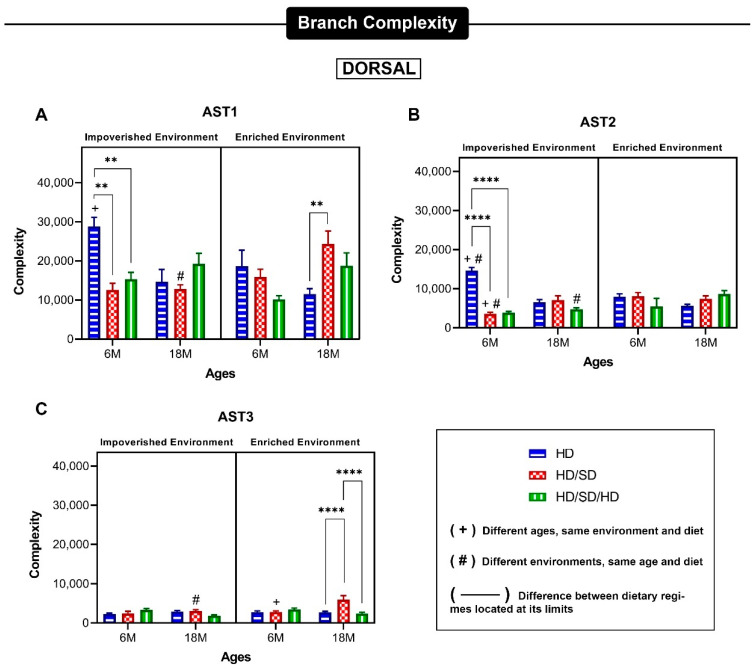
Mean values of astrocyte morphological phenotypes based on branch complexity at the external one-third of the molecular layer of the dorsal dentate gyrus. (**A**–**C**): Mean values of branch complexity and corresponding standard errors of astrocyte arbors to illustrate morphological differences between AST1 (**A**), AST2, (**B**) and AST 3 (**C**) for each experimental group based on impoverished environment and enriched environment, age (6M—6-month-old; 18M—18-month-old) and masticatory regimen (HD, HD/SD, and HD/SD/HD). HD: hard diet/pellet; SD: soft diet/powder food; ** *p* < 0.01 and **** *p* < 0.0001, by three-way ANOVA.

**Figure 7 ijms-23-06342-f007:**
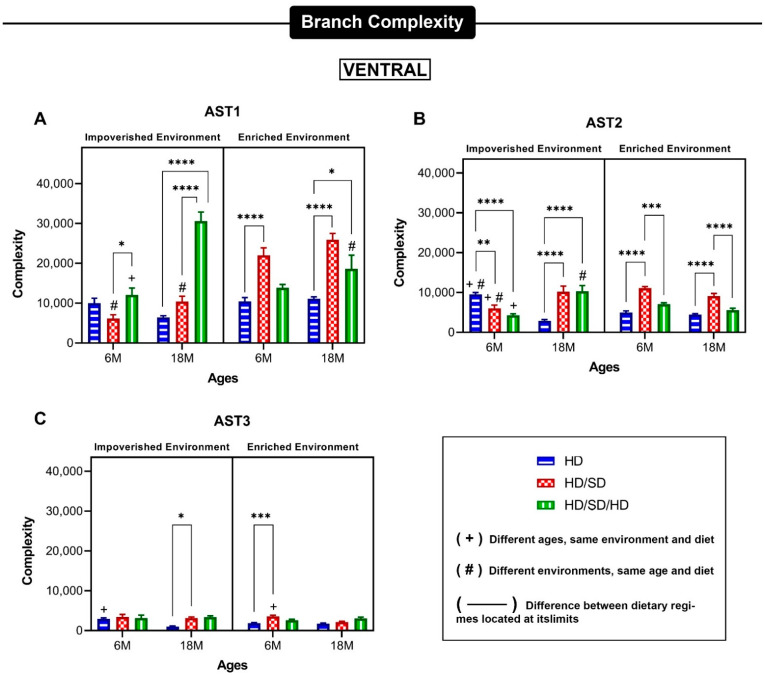
Mean values of morphological complexity for astrocytes morphological phenotypes from the external one-third of the molecular layer of the ventral dentate gyrus. (**A**–**C**): Mean values of branch complexity and corresponding standard errors of astrocyte arbors to illustrate morphological differences between AST1 (**A**), AST2 (**B**) and AST 3 (**C**) for each experimental group based on impoverished environment and enriched environment, age (6M—6-month-old; 18M—18-month-old) and masticatory regimen (HD, HD/SD, and HD/SD/HD). Asterisk (*) over connector bars denotes statistically significant difference (*) = *p* < 0.05; (**) = *p* < 0.01; (***) = *p* < 0.001; (****) = *p* < 0.0001.

## Data Availability

The authors will meet any demand allowing access to the required data.

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
