# Peer review of "The Sedentary Lifestyle and Masticatory Dysfunction: Time to Review the Contribution to Age-Associated Cognitive Decline and Astrocyte Morphotypes in the Dentate Gyrus"

_ijms, 2022, doi:10.3390/ijms23116342_

Round 1

Reviewer 1 Report

First of all, congratulations for the review on sedentary lifestyle associated with mastigatory function and environmental enrichment on the brain structure. The abstract is concise and shows the main points of the review that are punctuated in the text.

Some improvements can be made in the text:

  • Between lines 141 -174, the fluence of the text can be achieved citing some authors with similar points instead of studies cited separately.
  • In line 241, it is not clear the end of legend and the text.
  • In Figure 2 and 3 the size of lettering is too small. Moreover, the legend is not clear (differences between all diet regimes; differences between the ages; differences between the environments). For instance: if + indicates differences between ages in the same conditions (environment and diet) compared to 18M. Suggestion: (*) differences between diet regimes in the same age and environment; (+) difference between the ages in the same environment and diet; (#) difference between the environments in the same age and diet.
  • Figure 3E, the branch complexity (y-axis) can show only the scale 0 to 10000 to evidentiate the difference between the groups but I understand if the authors choose not to change in face of comparison with other astrocytes.

Author Response

Manuscript ID: ijms-1727785

THE SEDENTARY LIFESTYLE AND MASTICATORY DYSFUNCTION: TIME TO REVIEW THE CONTRIBUTION TO AGE-ASSOCIATED COGNITIVE DECLINE AND ASTROCYTE MORPHOTYPES IN THE DENTATE GYRUS

We would like to thank the reviewers' criticism and suggestions. They improved the present version of the manuscript.

As requested, the modifications were inserted along the text and are also described below.

Reviewer#1

Comments to the Author

“Between lines 141 -174, the fluence of the text can be achieved citing some authors with similar points instead of studies cited separately.”

Reply: As suggested, we reorganized the sentences of the paragraph hoping to improve the clarity of the text (lines 147-158): “…The elements inside enriched cages provide novelty, visuo-spatial and somatomotor stimuli and social interaction, but the stimuli for neurogenesis and the release of neurotrophins are originated from the voluntary exercise (Kobilo et al., 2011). Comparative effects of the elements provided by an enriched environment have enabled the disentanglement of the influence of novelty, social and physical activity and behavioral performance in hippocam-pal-dependent tasks. Indeed, well designed comparative studies demonstrated that running, but not a complex environment, stimulates hippocampal neurogenesis, whereas the complex environment, and not the running, increases depolarization-associated c-fos expression and reduces plasma corticosterone (Grégoire et al., 2014). However, the combination of cognitive stimuli, social interaction, and physical exercise was found to be the most effective way to reduce the neuropathological outcomes in a transgenic mouse model of cerebral amyloid angiopathy (Robison et al., 2020)…”

Comments to the Author

“In line 241, it is not clear the end of legend and the text.”

Reply: To distinguish the main text from the legends we now reduced the font size of all legends.

Comments to the Author

“In Figure 2 and 3 the size of lettering is too small. Moreover, the legend is not clear (differences between all diet regimes; differences between the ages; differences between the environments). For instance: if + indicates differences between ages in the same conditions (environment and diet) compared to 18M. Suggestion: (*) differences between diet regimes in the same age and environment; (+) difference between the ages in the same environment and diet; (#) difference between the environments in the same age and diet.”

Reply: The authors appreciate the suggestions, and all considerations were accepted, and adjustments were made in Figures 2 and 3.

Comments to the Author

“Figure 3E, the branch complexity (y-axis) can show only the scale 0 to 10000 to evidentiate the difference between the groups but I understand if the authors choose not to change in face of comparison with other astrocytes.”

Reply: In Figure 3E, the adoption of the same scale used in the graphic representation of Figure 3B allows us to readily visualize the differential impact of experimental variables on astrocytic morphotypes. Because of this facility provided by the use of the same scale, we decided to keep it as it is.

Reviewer 2 Report

The review paper addresses an important topic in research on the association between sedentary behavior and cognitive functioning. The paper is well written and the results of interest.

Some issues could be addressed before publication:

  1. I did not find information whether the Preferred Reporting Items for Systematic Reviews and Meta-Analyses (PRISMA) were followed.
  2. The practical implications could be elaborated in more detail.

What do the authors suggest for future research efforts?

What can we recommend to the public?

Which sub-populations may be particularly concerned, which not; why?

  1. The conceptual discussion could be expanded.

Which role does sedentary behavior play for cognitive aging in relation to other determinants such as socioeconomics, social relationships, etc.?

Author Response

Manuscript ID: ijms-1727785

THE SEDENTARY LIFESTYLE AND MASTICATORY DYSFUNCTION: TIME TO REVIEW THE CONTRIBUTION TO AGE-ASSOCIATED COGNITIVE DECLINE AND ASTROCYTE MORPHOTYPES IN THE DENTATE GYRUS

We would like to thank the reviewers' criticism and suggestions. They improved the present version of the manuscript.

As requested, the modifications were inserted along the text and are also described below.

Reviewer#2

Comments to the Author

“I did not find information whether the Preferred Reporting Items for Systematic Reviews and Meta-Analyses (PRISMA) were followed.”

Reply: The manuscript is a narrative review of the results published by us and by other authors and did not adopt the strategies of systematic review or meta-analysis. In fact, our data were revisited with an emphasis on the contribution of sedentary lifestyle and masticatory dysfunction on senile cognitive decline and the morphology of dentate gyrus astrocytes. For this purpose, we reanalyzed previous slide collections of published data set and showed new ways to interpret the results.

Comments to the Author

“The practical implications could be elaborated in more detail.”

Reply: Please let us know if the following paragraph from Concluding Remarks fulfil the Reviewer’s request: “…Overall, we have reviewed the rapidly expanding literature on astrocyte behavior in homeostatic conditions to highlight how the cellular and molecular characteristics relate to behavioral. In particular, we have focused on the cognitive decline aggravated by a sedentary lifestyle and by masticatory disorders by re-evaluating their influence on the morphology of the dentate gyrus astrocytes. We found that the reduction of masticatory activity reduced learning rates at all ages, which was independent of the environment, and that masticatory rehabilitation and environmental enrichment recovered age-related memory impairments. These spatial learning and memory deficits, and their recovery by an enriched environment and masticatory rehabilitation, were associated with differential morphological changes of the dentate gyrus astrocytes. Unbiased selection of astrocytes for three-dimensional reconstruction from the external one-third of molecular layer of dentate gyrus, using a random and systematic stereological approach followed by hierarchical cluster and discriminant analysis revealed the presence of four main astrocyte morphotypes. These morphological families identified in the external third of molecular layer of both dorsal and ventral dentate gyrus were differentially affected by age, sedentary lifestyle, and masticatory dysfunction suggesting, which suggests that they have different physiological roles in homeostatic conditions. All the morphotypes retained basic structural organization in intact tissue, suggesting that they are indeed subtypes of non-proliferative astrocyte profile. Future studies directly assessing astrocyte functions in the aging mouse brain should integrate molecular, cellular systemic and behavioral anal-ysis to unravel the underlying mechanisms associated with the aging cognitive decline aggravated by sedentary lifestyle and masticatory dysfunction…”

Comments to the Author

What do the authors suggest for future research efforts?

Reply: Considering the author’s conclusions in the present manuscript, being presented on the final period of the topic “Future studies directly integrate assessing astrocyte functions in the aging mouse brain should molecular, cellular systemic and behavioral analysis to unravel the underlying mechanisms associated with the aging cognitive decline aggravated by sedentary lifestyle and masticatory dysfunction”, the authors emphasize investing in molecular analysis, such as the transcriptome, and that the concentrations of certain hormones, such as cortisol, are investigated.

Comments to the Author

What can we recommend to the public?

Reply: It is important to encourage physical exercise, as well as other practices for purpose of somatomotor, visuo-spatial and cognitive stimulation as well as more attention must be devoted to oral health, especially for the elderly, in order to reduce the impacts on cognition and memory.

(Lauritano et al., 2019; Marchini et al., 2019; Scambler et al., 2021; Suzuki et al., 2021; Daly et al., 2018; Delwel et al., 2018; Miquel et al., 2018; Saito et al., 2018)

Comments to the Author

Which sub-populations may be particularly concerned, which not; why?

Reply: As suggested we now included (lines 61-66) the following paragraph on the main text. “… Because the damage on masticatory activity (Moreno et al., 2021; Nakamura et al., 2021; Suzuki et al., 2021; Xu et al., 2021; Chen et al., 2015; Tada & Miura, 2017; Lin, 2018; Miquel et al., 2018; Alvarenga et al., 2019; Kim et al., 2021), and the sedentary life style (Li et al., 2021; Ramírez-Rodríguez et al., 2014) are risk factors for age-related cognitive decline, we suggested special attention to the human sub-population facing greater damage to oral health or  stomatognathic system and long life sedentary life style…”.

Comments to the Author

The conceptual discussion could be expanded.

Reply: Because we have a variety of conceptual issues related to the review, we decided to discuss them throughout the text rather than concentrating them in one long paragraph. We hope that this way of doing it has not compromised the clarity of the text.

Comments to the Author

Which role does sedentary behavior play for cognitive aging in relation to other determinants such as socioeconomics, social relationships, etc.?

Reply: As our review focused on experimental data with animals, socioeconomic analysis and human social interaction were mentioned “en passant” but not reviewed.

It is important to note that the authors inserted Figures 2, 3 and 5 revised  in revised manuscript here attached, but did not uploaded them in a separate way .
